# Physical Activity and Emotional Regulation in Physical Education in Children Aged 12–14 Years and Its Relation with Practice Motives

**DOI:** 10.3390/healthcare11131826

**Published:** 2023-06-22

**Authors:** Jorge Rojo-Ramos, Juan Manuel Franco-García, Noelia Mayordomo-Pinilla, Francesco Pazzi, Carmen Galán-Arroyo

**Affiliations:** 1Physical Activity for Education, Performance and Health (PAEPH) Research Group, Faculty of Sports Sciences, University of Extremadura, 10003 Cáceres, Spain; 2Health Economy Motricity and Education (HEME), Faculty of Sport Science, University of Extremadura, 10003 Cáceres, Spain; jmfrancog@unex.es; 3Promoting a Healthy Society Research Group (PHeSO), Faculty of Sport Sciences, University of Extremadura, 10003 Cáceres, Spain; nmayordo@alumnos.unex.es (N.M.-P.); frapaz76@gmail.com (F.P.); 4Physical and Health Literacy and Health-Related Quality of Life (PHYQoL), Faculty of Sport Science, University of Extremadura, 10003 Cáceres, Spain; 5Sport, Health & Exercise Research Unit (SHERU), Department Sport and Well-Being, School of Education, Castelo Branco Polytechnic Institute, 6000-266 Castelo Branco, Portugal

**Keywords:** physical education, PLOC-2, secondary, sex, age, self-determined theory

## Abstract

This study aimed to analyze the different types of emotional regulation in first and second year high school students according to sex and age. Many adolescents do not meet the minimum WHO recommendations, at a critical stage in which habits that will later be maintained are established. For this reason, physical education is an important means to promote these habits and an understanding of the reasons for their participation in physical education. For this purpose, PLOC-2 was used. The Kolmogorov–Smirnov test was used to determine the characteristics of the data, the ANOVA test to explore the differences between sexes, and the Spearman test for correlations between the type of regulation and age. The results showed significant differences in several items and emotional regulation by sex and an inverse correlation between age and demotivation. There are differences between the reasons why both sexes perform physical activity, and we have determined that boys have more intrinsic regulation than girls do.

## 1. Introduction

Physical activity has been found to be a key factor in health development and disease prevention at all ages, improving quality of life, and as a tool for the motor development of children [1,2,3]. The World Health Organization (WHO) has established a series of physical activity recommendations according to each group to improve the quality of life of individuals and preserve their health as much as possible [4]. Spanish physical activity reports, specifically the Spanish National Health Survey (ENSE), report that 35.3% of the population aged between 15 and 69 years does not reach the minimum recommendations, with this lack being more frequent in women than in men [5]. Regarding children, 14% of them spend their free time in sedentary activities, with this being more frequent in girls than in boys. These data are alarming, since sedentary lifestyles are considered the fourth most important risk factor for mortality [6], and statistics indicate that the amount of leisure-time physical activity decreases in adolescence [7,8]. In addition, in the scientific literature, authors reported that childhood and pre-adolescence are periods in which character, personality, and life habits are formed, with a tendency to maintain these habits in the future [9,10]. In this sense, an association has been found between positive experiences in physical education classrooms and positive attitudes towards sports practice in later stages of education [11]; therefore, avoiding sedentary behaviors and creating healthy lifestyle habits in this period is essential to reduce the risk factors for mortality and the onset of various diseases.

In this sense, increasing the level of physical activity in children and adolescents is a priority. The area of physical education is the most suitable space for the development and promotion of these healthy habits, where we can increase time spent practicing movement, teach students how to exercise in their free time, and reduce their sedentary time, therefore increasing the possibility of their being active adults [10,12,13]. To achieve these objectives, it is of utmost importance that students become involved with the subject, showing a high level of interest in the subject and its contents, with a high motivation to achieve healthy lifestyle habits [14]. Motivation can be defined as the psychological perception of an activity that influences the cognitive, affective, and behavioral responses of the subject, guiding him/her to maintain that behavior [15]; therefore, proper motivation could facilitate the adoption of healthy habits in physical education classes [16,17].

In Spain, the education system is divided into primary education, with ages ranging from 6 to 12 years; compulsory secondary education (ESO), from 12 to 16 years of age; and Bachillerato, from 16 to 18 years of age. Physical education is compulsory in all educational cycles, except for Bachillerato, in which it is voluntary. In the present case, ESO is divided into two cycles: the first consists of the first three years, with students from 12 to 15 years of age, and the second consists of a single year, with students from 16 years of age. In the context of this article, physical education is a compulsory subject; however, teaching hours vary according to the autonomous community. In Extremadura, in the first cycle, two hours of physical education are taught per week, as in the second cycle [18]. The contents taught are all oriented to the promotion of healthy lifestyles, referring to physical activity, nutrition, psychological factors such as self-concept, self-esteem, and socialization, promoting positive values that integrate the inclusion of peers, and developing autonomy as far as possible. The specific contents in this first cycle are not delimited in a concise manner; that is, the educational curriculum states that students must have participated in at least one activity of each characteristic, such as cooperative, competitive, or invasive sports, but does not specify which ones must be performed during the academic year, giving teachers the freedom to choose the sports that best suit the students [19]. Self-determination theory (SDT) aims to explain the motivational states of people in different situations, in this case in physical education classes. This theory investigates the inherent growth and innate psychological needs of individuals to determine the basis for their development., establishing a continuum with different levels of self-determination and voluntariness from intrinsic motivation to amotivation [14,20]. This theory has different categories depending on voluntariness: if the subject performs physical activity on his/her own, it is self-determined motivation; if the activity is imposed, it is non-self-determined motivation [21,22]. Intrinsic motivation is the highest form of self-motivation, defined as the inherent tendency to seek novelty and explore new skills and improve and master them, and is a fundamental part of social and personal development. Within non-determined motivation, extrinsic motivation appears, which is more concerned with external factors than with those related to the task. Within this continuum, four other types of emotional regulation appear: integrated regulation, which appears when the activity is part of a habit; identified regulation, related to the benefits it brings to the subject; introjected regulation, when it is part of the subject, but is done to avoid internal punishments; and external regulation, related to the avoidance of punishment [23,24]. These dimensions are ordered from the highest to the lowest form of motivation. Finally, at the extreme end of the motivation spectrum is amotivation, which appears when a subject is neither intrinsically nor externally motivated [25].

To measure the magnitude of the different forms of motivational regulation, the Perceived Locus of Causality (PLOC) scale was created [26], which is used on different occasions in the context of physical education and is the most widely used tool, although this scale does not include the measurement of the integrated regulation of TAD. As a complement, other authors [27] modified this scale by including the items that correspond to integrated regulation, enunciated by Wilson et al. [28], so that information about this important emotional regulation could be extracted. Thus emerged the PLOC-2, a tool that will be applied in this paper. Understanding the behaviors of the different categories of motivation would reveal the reasons why students engage in physical activity and why they would continue to practice, facilitating teachers in designing strategies aimed at creating healthy habits by increasing motivation through methodologies applied with this objective in mind [25].

Based on this information, the aim of this study is to focus on the application of the PLOC-2 scale in schools in physical education classes, with the purpose of quantifying the different types of emotional regulation and the analysis of differences between sexes. As a secondary objective, the possible correlations between the variables of age and the different types of motivational regulation were also analyzed.

## 2. Materials and Methods

### 2.1. Participants

A total of 480 participants were included in this study. According to the Annual Report for the 2017–2018 school year available from the Regional Ministry of Education and Employment of the Regional Government of Extremadura: http://estadisticaeducativa.educarex.es/?alumnado/matricula/niveleducativo/ (accessed on 15 March 2023), the number (hereinafter, n) of students enrolled in compulsory secondary education was 43,477. From the sample, a Confidence Level (hereinafter, CN) of 95% was obtained, with a margin of error of ±4.5%. To determine the necessary sample size, a non-probabilistic, non-random sampling method based on convenience sampling was used [29]. Of the 480 students, 55.2% (N = 265) were girls, while the rest (44.8%, N = 215) were boys, achieving an equal sample with respect to the sex of the individuals. As for the school year, 53.1% (N = 255) belonged to the 1st year of ESO and the remaining 46.9% (N = 225) belonged to the 2nd year of ESO. The mean age of the students was 12.85 (0.69) years, their mean body mass index (BMI) was 19.06 (3.41) (kg weight/height in meters^2^) with a mean height of 1.59 m (0.07), and a mean weight of 48.36 kg (9.89). The sociodemographic data of the sample are in Table 1.

To participate in the study, a series of requirements were established for all the students: (1) Informed consent signed by their parents. (2) Taking the subject of PE in a public high school in the first cycle of compulsory secondary education (1st and 2nd years, ages between 12 and 14 years). (3) Knowledge of their height and weight.

The study complied with the ethical provisions of the Declaration of Helsinki and the protocol was approved by the Bioethics Committee of the University of Extremadura (Registration Code 71/2022).

### 2.2. Procedure

To obtain the contact data and the number of public school institutions compatible with the inclusion criteria of the study, the database of the Consejería de Educación y Empleo de la Junta de Extremadura (Department of Education and Employment of the Regional Government of Extremadura) was accessed.

Once the data were collected, physical education teachers were contacted by e-mail with a message containing the description of the study together with the objectives and the procedure, as well as the informed consent to be completed and signed by the parents and a model of the instrument to be used by the researchers on the students. If the teacher agreed to collaborate with the research team, a day would be established for the researchers to go to the school and carry out the intervention.

Once there, the researchers collected informed consents to ensure that those who were to participate met the inclusion criteria. After collecting them, each participant was provided with a tablet containing the form on the Google Forms platform and the researcher proceeded to read and explain the items on it, confirming that the students understood what was being asked.

An e-questionnaire was chosen because of its advantages regarding storing and processing the information and the media and paper it saves, in addition to the fact that its completion did not take more than 10 min.

### 2.3. Instruments

Sociodemographic data: for the characterization of the sample, a questionnaire on social and demographic characteristics was created consisting of five items (sex, age, grade, height, and weight). The body mass index was calculated using the mathematical formula designated for this purpose: BMI = weight in kilograms/(height in meters)^2^. The height and weight data were obtained from the information provided by the participants.

Physical Education Perceived Locus Of Causality-2 (PLOC-2): The PLOC-2 questionnaire designed and validated in Spanish was used, so no translation or cultural adaptation to the study population was carried out [27]. This scale begins with the sentence: “I participate in physical education classes” and then presents twenty-four items divided into four dimensions aimed at evaluating intrinsic motivation, integrated regulation, identified regulation, introjected regulation, external regulation, and amotivation. This instrument uses a Likert scale of 1–7 with 1: strongly disagree and 7: strongly agree. With respect to the psychometric properties of the instrument, the authors reported the following reliability values based on Cronbach’s alpha coefficient: α1 = 0.84; α2 = 0.93; α3 = 0.84; α4 = 0.69; α5 = 0.69; α6 = 0.82.

The table below shows the distribution of the 24 items of the instrument in each of the dimensions.

### 2.4. Statistical Analysis

For subsequent statistical procedures, it is necessary to determine the characteristics of the data, and for this purpose it was decided to apply the Kolmogorov–Smirnov test to determine whether the data met the assumption of normality. After application, the results showed that the data did not meet this assumption.

First, the scores obtained were analyzed according to the sex of the participants, exploring the differences between them using the ANOVA test. The significance level was determined as *p* < 0.05. To interpret the results, we established the intervals represented in the research articles [30]. A low correlation was determined to be indicated by values from 0.01 to 0.10, a medium correlation by values of 0.11 to 0.50, a strong correlation by values between 0.51 and 0.75; a correlation of 0.76 to 0.90 is considered high, and above 0.91 is considered a perfect correlation.

In order to determine the effect of the sex variable, Hedges’ g was applied, which has intervals that determine the magnitude of the effect. A magnitude below 0.20 is considered non-existent, from 0.21 to 0.49 is considered a small effect, from 0.50 to 0.80 is considered a moderate effect, and above 0.80 is considered a strong effect [31].

Finally, the Bonferroni correction for multiple comparisons was used to determine the *p*-value for the questionnaire items, resulting in *p* < 0.002.

## 3. Results

With the application of the Physical Education Perceived Locus of Causality-2 (PLOC-2) scale, differentiating by sex in the application of the ANOVA test, the results shown in Table 2 were obtained. 

In the Table 3, significant differences by sex appear in items 1, 2, 3, 8, 9, 10, 14, 18, 20, and 21.

Table 4 shows the results divided by dimensions according to the sex of the students. The ANOVA test was again applied to explore the differences between groups. Hedges’ *g* value was also included to analyze the magnitude of the effect of the sex variable. The results show significant differences in the dimensions (2) integrated regulation, (3) identified regulation, and (4) introjected regulation, with boys scoring higher than girls in the first two dimensions, and girls scoring higher than boys in dimension 4, introjected regulation.

Finally, Table 5 shows the Cronbach’s alpha indexes for each of the dimensions of the scale applied. The literature determines that values above 0.70 are considered satisfactory [32]. All dimensions obtained values above 0.72.

## 4. Discussion

### 4.1. Main Findings

This study analyzes the differences by sex in the different items of the PLOC-2 scale and in the different motivational dimensions of students in their first and second years of secondary school. Furthermore, the possible correlations between the dimensions and the variables of age and sex were studied. In the general results, significant differences by sex were observed for some items of the scale. In the dimensions, significant differences appeared in integrated, identified, and introjected regulations. Significant correlations were only found in the “amotivation” dimension with respect to age, of an inverse nature.

In the results obtained in the scale, differentiated by sex, significant differences were obtained in an element related to the first dimension, “intrinsic motivation,” with boys scoring higher than girls, although in the subsequent analysis of the types of motivation and regulation no significant differences were found in this regulation. In this type of motivation, the scientific literature, in general, reports higher scores for self-determined motivation in boys than in girls [33,34], in agreement with the results of this study, although no significant differences were found. This could mean that boys are more physically active and participate more in physical education because they enjoy physical activity more. Significant differences were obtained in all the items related to integrated regulation, with higher scores in boys than in girls; in the analysis by dimensions, significant differences were found, with boys having the highest scores. In this sense, the scientific literature has obtained similar results [35]. This could be due to the fact that, in general, boys tend to practice more physical activity in their daily lives, as reported by statistics, it being part of their daily routine [5]. In three of the four items related to identified regulation, significant differences were found, with the male gender scoring higher than the female sex. In the analysis of the general dimension, significant differences were obtained by sex, with boys scoring higher than girls. Some studies support these results [33], although others find no significant differences [36]. Although there were significant differences, the dimension “identified regulation” had the highest score in both sexes. This may mean that one of the main reasons students practice physical activity is because they know that physical activity is good for their health. In the next dimension, “introjected,” only one item reports significant differences, with girls scoring higher than boys; in the subsequent analysis by dimensions, this regulation had significant differences, with girls scoring higher than boys. In this sense, the scientific literature supports these results [34,35,37], reporting that the female sex has, in general, less interest in physical education classes, revealing that girls participate in physical education classes more to avoid feeling bad than for the benefits it can bring or for personal pride [38]. It is important for the female sex to improve its motivation profile and move towards intrinsic motivation [39], since low levels of motivation are associated with progressive physical activity abandonment [40,41,42]. Those with low motivation are often far from achieving the adoption of healthy lifestyle habits and, therefore, have a greater likelihood of developing diseases in the future. Finally, only one element showed no significant differences by sex, related to the dimension “amotivation.” although girls scored higher than boys. In this dimension, no significant differences were found, in line with what was found in the literature, although girls tend to have higher levels of amotivation in the subject [36,43]. Relative to Hedges’ *g* value, sex had a small effect on intrinsic, introjected, integrated, and identified regulation.

Regarding the correlation analysis between the types of motivation and the variable age, no significant correlations were found except in the “amotivation” dimension, where a medium inverse correlation appears, which means that the older the student, the lower the score in amotivation, showing increasing motivation. These are unusual results, since in most of the articles authors indicate an opposite relationship, where the older the students, the greater their amotivation towards physical education classes [44,45]. This can be explained by the homogeneity of the sample, since they were from two contiguous academic years, and age had little variability.

However, those students who are more motivated or unmotivated may be influenced by the sport being taught at the time of the researchers’ intervention, since the taste and interest of the students play a very important role in the development of motivation.

### 4.2. Practical Applications

The results of motivation questionnaires can provide information about the reasons why students engage in physical activity, and the difference between the different types of motivation can reveal the path that teachers should take. In this case, it would be interesting to apply methodologies that are more task-oriented [46,47], since they enhance intrinsic motivation and therefore increase the likelihood that they will adopt an active and healthy lifestyle that they can maintain in the future [48]. On the other hand, it would also be interesting to apply this type of methodology specifically for girls, since they participate more in EF by external regulation and therefore have less self-determined motivation than boys.

### 4.3. Limitations and Future Lines of Inquiry

This study has some limitations: (1) it is a cross-sectional study, so causal relationships cannot be established; (2) electronic questionnaires were used, and although they have certain advantages, they also have some disadvantages [49]; and (3) a non-probabilistic, non-random sampling method based on convenience sampling was used, so the results should be considered with caution. In addition, the sport played at the time of questionnaire implementation, which could have an influence on students’ motivation, was not considered. In the future, it would be interesting to extend the educational stages of the sample to obtain more representative data on age. It is also necessary to emphasize that the information on height and weight was self-reported, so that for those students who have complex emotions about their appearance, it is likely that they did not put their real weight or height, or perhaps did not know it and put an approximate one. Therefore, the BMI results should be interpreted with caution.

## 5. Conclusions

To summarize the results, the study shows that there are significant differences in the types of motivational regulation according to sex. Boys perform physical education because it is part of themselves, and they know the benefits they can obtain from it. It seems necessary to seek strategies to get the female sex to practice physical activity from a more self-determined approach to increase adherence to active and healthy habits in this sex through strategies that increase knowledge of the benefits of physical activity, in addition to implementing didactic units more adjusted to their tastes and needs. By contrast, an inverse correlation was observed between age and demotivation, which decreased with increasing age. Despite the differences between sexes, it is advisable to include educational strategies that improve and increase the intrinsic motivation of all students, as most students practice extrinsic motivation.

## Figures and Tables

**Table 1 healthcare-11-01826-t001:** Sample characterization (n = 480).

Variable	Categories	N	%
Sex	Boys	215	44.8
	Girls	265	55.2
Grade	1st E.S.O.	255	53.1
	2nd E.S.O.	225	46.9
**Variable**		**M**	**SD**
Age (years)		12.85	0.69
BMI (kg/m^2^)		19.06	3.41
Height (in meters (m))		1.59	0.07
Weight (in kilos (kg))		48.36	9.89

N: number; %: percentage; SD: standard deviation; M: Mean. Note: E.S.O. = Education Secondary Obligatory; BMI: body mass index; m: meters; kg: kilograms.

**Table 2 healthcare-11-01826-t002:** Distribution of items according to the dimensions.

Dimension	Items
Intrinsic motivation	1, 7, 13, 19
Integrated regulation	2, 8, 14, 20
Identified regulation	3, 9, 15, 21
Introjected regulation	4, 10, 16, 22
External regulation	5, 11, 17, 23
Amotivation	6, 12, 18, 24

**Table 3 healthcare-11-01826-t003:** Scores obtained according to sex and specialty in each of the items of the instrument.

Item	Sex		
	Boys	Girls	
I Participate in Physical Education Classes …	M (SD)	M (SD)	*p*
1. Because Physical Education is fun.	6.76 (0.47)	6.39 (1.16)	<0.001 *
2. Because it agrees with my lifestyle.	6.14 (1.07)	5.45 (1.52)	<0.001 *
3. Because I want to learn sports skills.	6.40 (1.10)	5.98 (1.34)	<0.001 *
4. Because I want the teacher to think I am a good student.	4.47 (2.12)	4.84 (1.79)	0.038
5. Because I’ll be in trouble if I don’t.	3.91 (2.17)	3.74 (2.13)	0.386
6. But I don’t really know why.	2.19 (1.83)	2.23 (1.86)	0.812
7. Because I enjoy learning new skills.	6.20 (1.09)	6.09 (1.33)	0.334
8. Because I consider physical education a part of me.	5.79 (1.58)	5.15 (1.73)	<0.001 *
9. Because it is important to me to do well in Physical Education.	6.12 (1.22)	5.68 (1.35)	<0.001 *
10. Because I would feel bad about myself if I didn’t.	3.93 (2.16)	5.04 (1.92)	<0.001 *
11. Because that’s what I’m supposed to do.	4.84 (2.02)	4.57 (1.86)	0.128
12. But I don’t understand why we should have Physical Education.	1.49 (1.24)	1.30 (0.57)	0.031
13. Because Physical Education is stimulating.	5.67 (1.33)	5.34 (1.58)	0.015
14. Because I see Physical Education as a fundamental part of who I am.	5.44 (1.81)	4.77 (1.84)	<0.001 *
15. Because I want to improve in sports.	6.21 (0.98)	6.04 (1.41)	0.132
16. Because I want the other students to think I am skilled.	3.95 (1.98)	3.90 (2.15)	0.787
17. So that the teacher does not yell at me.	2.05 (1.74)	2.11 (1.72)	0.675
18. But I really feel like I’m wasting my time in PE.	1.42 (1.28)	1.64 (1.16)	0.047
19. For the satisfaction I feel while learning new skills/techniques.	5.86 (1.41)	5.62 (1.77)	0.111
20. Because I consider that Physical Education agrees with my values.	6.00 (1.18)	5.47 (1.63)	0.001 *
21. Because I can learn skills that I could use in other areas of my life.	6.26 (1.16)	5.85 (1.46)	0.001
22. Because I worry when I don’t.	3.93 (1.96)	4.13 (1.80)	0.242
23. Because that is the norm.	3.77 (2.17)	3.25 (1.75)	0.004
24. But I can’t understand what I’m getting out of Physical Education.	2.12 (1.64)	1.92 (1.49)	0.183

Note: *p* is significant < 0.002 *. M = mean value; SD = standard deviation. Each score obtained is based on a Likert scale (1–7): 1 “Strongly disagree” and 7 “Strongly agree”.

**Table 4 healthcare-11-01826-t004:** Descriptive analysis, differences, and effect size of the PLOC-2 dimensions.

Dimension	Sex			
	Boys	Girls		
	M (SD)	M (SD)	*p*	*g*
1. Intrinsic motivation	6.12 (0.78)	5.86 (1.25)	0.008	0.243
2. Integrated regulation	5.84 (1.19)	5.21 (1.50)	<0.001 *	0.457
3. Identified regulation	6.24 (0.82)	5.88 (1.07)	<0.001 *	0.367
4. Introjected regulation	4.06 (1.62)	4.47 (1.42)	0.004	0.269
5. External regulation	3.63 (1.60)	3.41 (1.43)	0.107	0.148
6. Amotivation	1.80 (1.20)	1.77 (1.02)	0.777	0.026

M = media; SD = standard deviation. Each score obtained is based on a Likert scale (1–7). * *p* is significant < 0.05. There is a small effect when *g* > 0.21, a medium effect *g* > 0.5 and a large effect *g* > 0.8.

**Table 5 healthcare-11-01826-t005:** Cronbach’s alpha coefficients for each dimension.

Dimension	Cronbach’s Alpha
Intrinsic motivation	0.81
Integrated regulation	0.91
Identified regulation	0.72
Introjected regulation	0.75
External regulation	0.78
Amotivation	0.77

## Data Availability

The datasets used during the current study are available from thecorresponding author upon reasonable request.

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
