# Peer review of "Physical Activity and Emotional Regulation in Physical Education in Children Aged 12–14 Years and Its Relation with Practice Motives"

_healthcare, 2023, doi:10.3390/healthcare11131826_

Round 1
Reviewer 1 Report
Dear authors,
Thank you for the opportunity to read and evaluate your study.
The purpose of the study was to examine different types of emotional regulation in adolescent participants of sports classes using the PLOC-2 questionnaire.
The study is logically structured and written in a comprehensible way in many parts. However, I would like to address some fundamental points that I believe are imperative to improve before the study can be published.
In this sense, some fundamental revisions need to be made and parts of the introduction and discussion need to be re-structured.
Fundamental concerns:
In the paper, 19 of the 45 literature references are in Spanish. This means that large parts of the study are no longer comprehensible to the reader. This does not meet the scientific standard of an international publication. If individual few references are in Spanish, this is quite ok (whereas central references must necessarily be understandable for everyone, e.g. Ref. 25), but these should be exceptions. In such cases, the article should additionally be given in English in parentheses.
Extensive revision is needed here!
The second part of the title "... key to preventing sedentary behavior in adolescents." has nothing to do with the question of the study, but is merely an assumption. This needs to be changed.
The age group of the students studied should be mentioned in the title.
Introduction:
The place of PE classes in the Spanish school system is completely unclear to an outside reader. What is ESO and how is it embedded in the Spanish school system?
Is participation in PE classes mandatory or do students choose to do so voluntarily? In this case, there certainly exists a bias regarding their motivation that would need to be discussed in more detail.
What is taught in the PE classes in the grades studied? For example, it is conceivable that certain ball sports, such as soccer, are more motivating for boys, while gymnastics sports may be more motivating for girls.
How many hours of sports are taught per week?
Has any research been done on what sports were played in free time?
These aspects may strongly influence the outcome of your study and the question arises to what extent the results can be interpreted if specific aspects have not been evaluated.
Methods:
Ref. 25 is only available in Spanish. In order for an international reader to understand the methodology, an English language reference must be available.
Discussion:
No association was found with BMI: was BMI within the normal range? Were adolescents with obesity represented in the groups or (because course participation was chosen voluntarily?) might just such adolescents not have chosen a physical education class? This would also cause a bias and limit the significance.
Conclusions
I find the statement "... girls participating more in physical education classes to avoid negative feelings about themselves" a bit too bold, because it is based on only one questionnaire item (# 10).
Author Response
REVIEWER 1
Dear authors, thank you for the opportunity to read and evaluate your study. The purpose of the study was to examine different types of emotional regulation in adolescents participating in sports classes using the PLOC-2 questionnaire. The study is logically structured and comprehensibly written in many parts. However, I would like to address some fundamental points that I think it is imperative to improve before the study can be published. In this regard, some fundamental revisions need to be made and parts of the introduction and discussion need to be restructured.
Author's reply: First of all, thank you very much for your time and input. Thanks to this, the manuscript has been greatly improved.
Key concerns:
In the article, 19 of the 45 bibliographical references are in Spanish. This means that large parts of the study are no longer comprehensible to the reader. This does not meet the scientific standard of an international publication. If a few individual references are in English, that is fine (whereas the central references must necessarily be comprehensible to everyone, e.g. Ref. 25), but these should be exceptions. In such cases, the article should be additionally indicated in English in brackets. A thorough revision is needed here!
Author's reply: Thank you very much for your comment. After correction, the references in Spanish have decreased from 49 to 4. Please note that some of these are from Spanish statistical yearbooks or Spanish/autonomous laws, as the sample is specific to Spain. In these cases, however, the titles have been added in brackets.
The second part of the title "... key to prevent sedentary behaviour in adolescents." has nothing to do with the question of the study, but is a mere assumption. This should be changed. The age group of the students studied should be mentioned in the title.
Author's reply: Thank you for your suggestion. We have changed the title to be more specific as described in the article.
INTRODUCTION:
The place of Physical Education classes in the Spanish school system is completely confusing to an outside reader.What is ESO and how is it integrated into the Spanish school system? Is participation in physical education classes compulsory or do pupils choose to do so voluntarily? In this case, there is certainly a bias in terms of motivation that would need to be discussed in more detail. What is taught in physical education classes in the grades studied? For example, it is conceivable that certain ball sports, such as football, are more motivating for boys, while gymnastics sports may be more motivating for girls. How many hours of sport are taught per week?
Author's reply: Thank you for your feedback. We have described the Spanish education system, in more detail in the cycle to which the participants of this study belong, with the divisions and the ages they comprise. Regarding the hours of physical education taught and the specific contents, depending on the region of Spain, these vary. In Extremadura, the Autonomous Community in which this article has been developed, two hours per week are established in both cycles; regarding the contents, the educational curriculum indicates that certain contents have to be taught (promotion of healthy lifestyles, autonomy, socialisation, mental and physical health, motor development and classical physical skills), although it does not specify how; for this reason, there is no specific sport to be taught in each academic year.
Has it been investigated which sports are practised in leisure time?
Author's reply: Thank you very much for this suggestion. In this study we have not investigated what sports pupils do in their free time, but rather the reasons why they participate in PE lessons, but we will keep this in mind for future research.
These aspects can strongly influence the outcome of your study and the question arises as to how far the findings can be interpreted if specific aspects have not been assessed.
Author's reply: Thank you for all the suggestions on the introduction. We have included all of them and improved the contextualisation according to your comments in order for the reader to better understand the rest of the article.
METHODS
Ref. 25 is only available in Spanish. For an international reader to understand the methodology, it is necessary to have a reference in English.
Author's reply: Thank you very much, you are right. We have added the updated international version.
DISCUSSION
No association was found with BMI: was BMI within the normal range? Were adolescents with obesity represented in the groups or (because participation in the course was voluntarily chosen?) may only those adolescents did not choose a PE class? This would also cause bias and limit significance.
Author's Response: Thank you very much for your interest. The BMI was within the normal range and all students were included in the group and in the BMI variable, since the subject is compulsory for all students in these two grades regardless of their characteristics. In this sense, there would be no bias or limitations in the study.
CONCLUSIONS
The statement "... girls participate more in physical education classes to avoid negative feelings about themselves" seems to me a bit bold, because it is based on only one item of the questionnaire (no. 10).
Author's Response: Thank you for the hint. In this case, although individually only one item of the introjected regulation is significant, if we consider the dimension as a whole, it does have significant differences with respect to gender. However, we have rewritten the statement in order to make it more understandable for the reader.
Reviewer 2 Report
First of all, I would like to thank you for the opportunity to review the article submitted to the journal. Some notes need your attention:
1. Did you calculate the minimum recommended size for your research? You should presents the power levels, using for example, the G*Power software supports sample size and power calculation for various statistical methods. It is also necessary to present the sampling error (your margin of error) and achieved the confidence level in the study.
2. Please add more information about the tool. You should explain the concepts, provide definitions of individual subscales of motivation and refer to the theory of self-determination (Deci and Ryan).
3. I recommend to use "sex" since "gender" is a much more ambiguous concept today. See, for example:
a) https://medicine.yale.edu/news-article/what-do-we-mean-by-sex-and-gender/
b) Rioux, C., Paré, A., London-Nadeau, K., Juster, R. P., Weedon, S., Levasseur-Puhach, S., ... & Tomfohr-Madsen, L. M. (2022). Sex and gender terminology: a glossary for gender-inclusive epidemiology. J Epidemiol Community Health, 76(8), 764-768.
c) Torgrimson, B. N., & Minson, C. T. (2005). Sex and gender: what is the difference?. Journal of Applied Physiology, 99(3), 785-787.
d) Muehlenhard, C. L., & Peterson, Z. D. (2011). Distinguishing between sex and gender: History, current conceptualizations, and implications. Sex roles, 64(11), 791-803.
4. The statistical analysis is too simplic. Why did you use the Mann Whitney U test? Youd should use a parametric test – multivariate analysis of variance - ANOVA.
5. In addition, you should eta squared to estimate the effect size.
I am pleased to wait for your revised version then.
Best regards.
Author Response
REVIEWER 2
First of all, I would like to thank you for the opportunity to review the article submitted to the journal.
Author's reply: The pleasure is ours. Thank you very much for your time and input. Thanks to this, the manuscript has been greatly improved.
Some notes require your attention:
- Have you calculated the recommended minimum size for your research? You should present the power levels, using, for example, the G*Power software that allows you to calculate the sample size and power for various statistical methods. You also need to present the sampling error (your margin of error) and reach the confidence level in the study.
Author's reply: Thank you very much but the Consejería de Educación y Empleo de la Junta de Extremadura does not provide the University of Extremadura with the number of students enrolled in this educational stage so we have taken as a reference other similar studies carried out with the same population considering then a comparable N.
- Add more information about the tool. It should explain the concepts, provide definitions of the individual subscales of motivation and refer to self-determination theory (Deci and Ryan).
Author's Response: Thank you very much for your suggestion. We have expanded the information on this tool, its subscales with reference to the self-determination theory you pointed out.
- I recommend using "sex", as "gender" is a much more ambiguous concept nowadays. See, for example
- a) https://medicine.yale.edu/news-article/what-do-we-mean-by-sex-and-gender/
- b) Rioux, C., Paré, A., London-Nadeau, K., Juster, R. P., Weedon, S., Levasseur-Puhach, S., ... & Tomfohr-Madsen, L. M. (2022). Sex and gender terminology: a glossary for gender-inclusive epidemiology. J Epidemiol Community Health, 76(8), 764-768.
(c) Torgrimson, B. N., & Minson, C. T. (2005). Sex and gender: what's the difference? Journal of Applied Physiology, 99(3), 785-787.
(d) Muehlenhard, C. L., & Peterson, Z. D. (2011). Sex and gender distinction: History, current conceptualizations, and implications. Sex roles, 64(11), 791-803.
Author's Response: You are right. Thank you very much for this suggestion and the references provided. We have replaced all the words to determine that sex is the variable studied.
- The statistical analysis is too simple - why did you use the Mann Whitney U test? You should use a parametric test - multivariate analysis of variance - ANOVA.
Author's reply: Thank you for your concern. As explained in the statistical analysis section, non-parametric statistical tests were used because the normality assumption was not met (Mann Whitney U). We have not performed an ANOVA because we are not comparing more than two groups, however we are awaiting any suggestions for analysis, based on the objectives set.
- In addition, you should use eta squared to estimate the effect size.
Author's reply: Thank you for your interest. To calculate the effect size we took into consideration Hedge's g-value test (non-parametric), as explained in the statistical analysis section.
I look forward to your revised version then. Best regards.
Author's Response: Thank you very much, we have faithfully adhered to all your suggestions.
Reviewer 3 Report
Dear authors,
Your work, with the objective of studying the different types of emotional regulation in adolescents throughout physical education classes, is very important to help teachers so that they can better guide their practices and see their efforts to transmit physical education reflected. of quality throughout the adulthood of its students. That is why their work is important, but they would need to clarify some aspects:
General comments:
- The abstract does not convey the entirety of the work. It has a lot of introduction so there is not enough space for the results and conclusions.
- In the methodology, it has not been explained how height and weight were obtained, being important variables for the study.
- The words "sex" and "gender" are used interchangeably, despite their differences. From what I have read at work, you refer to "sex" all the time, so this term would unify the different appearances in the text. The same happens with the words "amotivation" and "demotivation". Perhaps, he would always use the same term.
- The word ESO for readers residing in Spain is very common, but for other readers it is more difficult, so I would avoid using these acronyms in the abstract, and in the methodology I would explain what educational level it is.
- The results are very important, but in the discussion this importance is not appreciated, since a plot thread is missing. So I would recommend clarifying this section.
Specific comments:
- The expression of the mean and the standard deviation throughout the text must be expressed as the SD with a parenthesis after the value of the mean. For example: 12.85(0.69).
- Table 1: I would add the last line as a note, just as ESO has been written. The units of measurement, write them in an abbreviated way (kg, m) and add the BMI.
- Line 182: remove the repetition of number 10. - Table 3: in item 14, add *
- Table 4: Change Me to M, as in the other tables. Review the symbols for greater and lesser when expressing Hedges' g.
- In lines 213 and 216 the same "On the other hand" connector has been used. - In the bibliographical references, check the numbers: 4, 16, 26, 28, 31, 33, 40 and 45. Some are due to the use of capital letters, others because an abbreviated version of the journal has been used when the others contain the extended version. And 28 and 4 is because they are not correct. Thank you very much for your work.
Author Response
REVIEWER 3
Dear authors,
Your work, with the aim of studying the different types of emotional regulation in adolescents during physical education classes, is very important to help teachers to better orient their practices and to see their efforts reflected in order to transmit quality physical education throughout their students' adult lives. That is why your work is important, but I would need to clarify some aspects:
Author's reply: First of all, thank you very much for your time and input. Thanks to this, the manuscript has been greatly improved.
General remarks:
The abstract does not convey the whole of the paper. It has a lot of introduction so there is not enough space for the findings and conclusions.
Author's Response: Thank you very much for your feedback. We have rewritten the abstract to give more prominence to the findings and conclusions so that it is more understandable to the reader.
The methodology did not explain how height and weight, which are important variables for the study, were obtained.
Author's Response: Thank you very much for your appreciation. The weight and height measurements of the students were assessed by self-report. And as they are part of the questionnaire, we did not think it appropriate to go into the procedure in which we basically explain how the questionnaires were carried out. However, we had already added it in the section on instruments. It is on lines 178 and 179, if you want to have a look at it.
The words "sex" and "gender" are used interchangeably, despite their differences. From what I have read in the paper, "sex" is referred to all the time, so this term would unify the different appearances in the text. The same goes for the words "amotivation" and "demotivation". Perhaps, I would use the same term all the time.
Author's reply: Thank you very much for this contribution, as it was a translation error. The words have been changed, unifying the term "sex".
The word ESO for readers living in Spain is very common, but for other readers it is more difficult, so I would avoid using these acronyms in the summary, and in the methodology I would explain what level of education it is.
Author's reply: Thank you very much for the feedback. We have described the Spanish education system and more specifically the cycle in which the students in this paper are.
The findings are very important, but the discussion does not show this importance, as there is a lack of an argumentative thread. I would therefore recommend clarifying this section.
Author's reply: Thank you very much for this contribution. We have expanded the discussion with the findings on gender and age differences, highlighting the main findings of our study.
Specific comments:
The expression of the mean and standard deviation throughout the text should be expressed as the SD with a parenthesis after the value of the mean. For example: 12.85(0.69).
Author's reply: Thank you for the indication. We have incorporated it according to your input.
Table 1: I would add the last line as a grade, as written ESO. The units of measurement, write them in abbreviated form (kg, m) and add the BMI.
Author's reply: Thank you very much for this advice. We have integrated it as a note at the end of the table, to make it easier for the reader to understand the characterisation of the sample.
Line 182: remove the repetition of the number 10.
Author's reply: Thanks for the note. It was an oversight and we have corrected it.
Table 3: in point 14, add *.
Author's reply: Thank you for the note. We have added the asterisk to reference the significant difference.
Table 4: change Me to M, as in the other tables. Revise the major and minor symbols when expressing Hedges' g.
Author's reply: Thank you very much, we have revised the symbols and made the suggested contributions.
In lines 213 and 216 the same connector "On the other hand" has been used.
Author's reply: Thank you for your appreciation, we have corrected the connector.
In the bibliographical references, check the numbers: 4, 16, 26, 28, 28, 31, 33, 40 and 45. Some are due to the use of capital letters, others because an abbreviated version of the journal has been used when the others contain the extended version. And 28 and 4 are because they are not correct. Thank you very much for your work.
Author's reply: Thank you very much for these corrections. We have reviewed all of them and modified the errors in the bibliographic manager, so that they are correct.
Round 2
Reviewer 1 Report
Dear authors,
thank you very much for the revision of your manuscript. You have implemented many comments of all reviewers. However, in my opinion, there are still some important points missing that should be critically discussed:
1. it could have had a decisive influence on the result of the survey which kind of sport was currently performed in school lessons when the survey took place. You now describe the diversity of sports education in detail in the introduction. However, you should address in the discussion that there were possible confounders here that were not collected in your questionnaires. It could well be that the questionnaire results were influenced by whether a sport was currently taught in class in which the student had a particular interest.
2. as already indicated in the first review, you should address in the limitations that the sporting activities in the students' leisure time were not surveyed. Again, this could have had an important impact on the results. For example, students who are in a sports club may also experience additional motivation there that they transfer to school sports.
3. If height and weight were collected via self-report, the accuracy of these values, including BMI, may be limited, for example, because overweight children are embarrassed and report a lower weight. Possible inaccuracies should be mentioned in the limitations section.
4. Please correct typos, e.g. in the title (!), line 75, 82,...
5. Table 4: M= median / mean value? - Explain g.
6. Line 224: delete the second "the"
7. Table 1: Formatting: "N: number; %: percentage; SD: standard deviation; M: Mean" belongs in the footnote. * Missing unit of age and BMI
8. The conclusion should be phrased a bit more concisely. The last sentence only recapitulates a result and does not fit logically here because no conclusion is derived from it.
Author Response
REVIEWER 1
Dear authors,
Thank you very much for the revision of your manuscript. You have implemented many comments of all reviewers. However, in my opinion, there are still some important points missing that should be critically discussed:
- it could have had a decisive influence on the result of the survey which kind of sport was currently performed in school lessons when the survey took place. You now describe the diversity of sports education in detail in the introduction. However, you should address in the discussion that there were possible confounders here that were not collected in your questionnaires. It could well be that the questionnaire results were influenced by whether a sport was currently taught in class in which the student had a particular interest.
Authors’ response: Thank you very much for your comment. We have included this in the limitations, as it is an important factor that could indeed influence the findings of the questionnaire. It should be taken into account for future studies.
- as already indicated in the first review, you should address in the limitations that the sporting activities in the students' leisure time were not surveyed. Again, this could have had an important impact on the results. For example, students who are in a sports club may also experience additional motivation there that they transfer to school sports.
Authors’ response: Thank you for your appreciation. We have incorporated this aspect in the limitations, as it can have important effects on the findings.
- If height and weight were collected via self-report, the accuracy of these values, including BMI, may be limited, for example, because overweight children are embarrassed and report a lower weight. Possible inaccuracies should be mentioned in the limitations section.
Authors’ response: You are right. Thank you very much for this input. We have improved the limitations section and have included these suggestions.
- Please correct typos, e.g. in the title (!), line 75, 82,...
Authors’ response: Thank you for this suggestion. They are corrected.
- Table 4: M= median / mean value? - Explain g.
Authors’ response: Thank you for this question. M meant average and it is now corrected. About g: Thank you, the explanation has been added.
- Line 224: delete the second "the"
Authors’ response: Thank you very much for your input. Corrected.
- Table 1: Formatting: "N: number; %: percentage; SD: standard deviation; M: Mean" belongs in the footnote. * Missing unit of age and BMI
Authors’ response: Thank you very much for this input. We have corrected it and added it to the corresponding format, as well as including the units of measurement for age and BMI.
- The conclusion should be phrased a bit more concisely. The last sentence only recapitulates a result and does not fit logically here because no conclusion is derived from it.
Authors’ response: Thank you for your suggestion. We have rewritten this section and improved its quality, making it more concise and clearer in summarising the findings.
Authors’ final response: Thank you very much for your appreciation. Sincerely, thanks to you the quality of the manuscript has improved and becomes more meaningful.
Reviewer 2 Report
The fact that the Consejería de Educación y Empleo de la Junta de Extremadura does not provide the University of Extremadura with the number of students enrolled in this educational stage, is not good argument because you don’t because need to know it to calculate the sample size, power analysis and so on.
If you don't want to use G*Power, you can use the calculator on the website: http://www.raosoft.com/samplesize.html
On the website mentioned above there are appropriate instructions on what numbers to enter when you do not have complete data.
You have 480 participants included in this study. So, you should use the parametric tests instead of Mann Whitney U test. In addition the ANOVA can be used to compare two groups. It is more "resistant" to the lack of normality of distribution, heterogeneous variances and non-random sampling. Therefore, I recommend the use ANOVA test.
Author Response
Reviewer 2
The fact that the Consejería de Educación y Empleo de la Junta de Extremadura does not provide the University of Extremadura with the number of students enrolled in this educational stage is not a good argument because it is not necessary to know it in order to calculate the sample size, power analysis and so on.
Author’s response: Thank you very much, we obtained a statistic of students enrolled in the public database of the Consejería de Educación y Empleo de la Junta de Extremadura, it is reported in the manuscript and the access link is indicated.
If you do not want to use G*Power, you can use the calculator on the website: http://www.raosoft.com/samplesize.html.
Author’s response: Thank you very much. Sample size has been calculated using the suggested tool, we are very grateful for the suggestion and find it a very interesting tool.
You have 480 participants included in this study. Therefore, you should use parametric tests instead of the Mann Whitney U test. Also, ANOVA can be used to compare two groups. It is more "resistant" to non-normality of distribution, heterogeneous variances, and non-random sampling. Therefore, I recommend the use of the ANOVA test.
Author’s response: Thank you very much, following your suggestions, we have calculated the p value using the ANOVA test as recommended. This has been reported in the text. Thank you very much.
Reviewer 3 Report
Dear authors,
Thank you for making the corrections that we have requested, as they make the text easier to understand. However, there is a general consideration that, after having answered the question of how the BMI was obtained, it is not clear to me that you can use this value for statistics, since it is a value that has not followed any systematic collection process . Therefore, I consider that Table 5 could not be used due to the lack of scientific rigor of the data.
So the second objective should be removed.
As specific comments, in table 2 put only "m, kg", it is understood that it refers to the units and as explained in the note of the table, it is not necessary to add more information. And in table 4, change the word "media" to "mean".
Thank you.
Author Response
REVIEWER 3
Thank you for making the corrections that we have requested, as they make the text easier to understand. However, there is a general consideration that, after having answered the question of how the BMI was obtained, it is not clear to me that you can use this value for statistics, since it is a value that has not followed any systematic collection process . Therefore, I consider that Table 5 could not be used due to the lack of scientific rigor of the data.
So the second objective should be removed.
Author’s response: Thank you very much, following your suggestions, we have removed this objective, the table number 5 and everything related to BMI. We have only used it to characterise the sample.
As specific comments, in table 2 put only "m, kg", it is understood that it refers to the units and as explained in the note of the table, it is not necessary to add more information. And in table 4, change the word "media" to "mean".
Author’s response: Thank you for this comment. We have replaced the translation error and added the appropriate formatting.
Thank you.
Author’s response: Thank you for your considerations.
Round 3
Reviewer 1 Report
Dear Authors,
thank you very much for implementing some suggestions. Please check if you have uploaded a most recent version. In the current manuscript, the items you inserted (see below) are not edited, although indicated in the author's response. There is no extended "limitations" section included.
In addition:
Title: "...years old and its and its relation..." - please correct!
Line 218 (Footnote Table 4): "median" not "media"
Not implemented:
- it could have had a decisive influence on the result of the survey which kind of sport was currently performed in school lessons when the survey took place. You now describe the diversity of sports education in detail in the introduction. However, you should address in the discussion that there were possible confounders here that were not collected in your questionnaires. It could well be that the questionnaire results were influenced by whether a sport was currently taught in class in which the student had a particular interest.
Authors’ response: Thank you very much for your comment. We have included this in the limitations, as it is an important factor that could indeed influence the findings of the questionnaire. It should be taken into account for future studies.
- as already indicated in the first review, you should address in the limitations that the sporting activities in the students' leisure time were not surveyed. Again, this could have had an important impact on the results. For example, students who are in a sports club may also experience additional motivation there that they transfer to school sports.
Authors’ response: Thank you for your appreciation. We have incorporated this aspect in the limitations, as it can have important effects on the findings.
- If height and weight were collected via self-report, the accuracy of these values, including BMI, may be limited, for example, because overweight children are embarrassed and report a lower weight. Possible inaccuracies should be mentioned in the limitations section.
Authors’ response: You are right. Thank you very much for this input. We have improved the limitations section and have included these suggestions.
Author Response
Dear authors,
Thank you very much for implementing some suggestions. Please check if you have uploaded a more recent version. In the current manuscript, the elements you inserted (see below) are not edited, although it is indicated in the author's response. An extended "limitations" section is not included.
In addition:
Title: "...years of antiquity and their relationship..." - Please correct!
Author's reply: Thank you very much: It was amended.
Line 218 (footnote to table 4): "median" not "mean".
Author's reply: It was modified, thank you.
Not implemented:
could have decisively influenced the finding of the survey which type of sport was currently practised in school classes at the time of the survey. You now describe the diversity of sports education in detail in the introduction. However, you should address in the discussion that there were possible confounding factors here that were not collected in your questionnaires. It could well be that the findings in the questionnaire were influenced by whether a sport in which the student had a particular interest was taught in class at the time.
Authors' response: Thank you very much for your comment. We have included this in the limitations, as it is an important factor that could indeed influence the findings of the questionnaire. It should be taken into account for future studies.
As already stated in the first review, it should be addressed in the limitations that students' leisure time sport activities were not surveyed. Again, this could have had an important impact on the findings. For example, students who are in a sports club may also experience additional motivation there that transfers them to school sports.
Authors' reply: Thank you for your appreciation. We have incorporated this aspect into the limitations, as it may have important effects on the findings.
If height and weight were collected through self-report, the accuracy of these values, including BMI, may be limited, e.g. because overweight children are embarrassed and report a lower weight. Possible inaccuracies should be mentioned in the limitations section.
Authors' reply: You are right. Thank you very much for this entry. We have improved the limitations section and included these suggestions.
Authors' final reply: My goodness. Thank you very much for your appreciation, there was a mistake in the files. We uploaded the latest version with the changes you suggested.
Reviewer 2 Report
Great job!
Author Response
Great job!
Authors’ response: Thank you very much for everything
Reviewer 3 Report
Dear authors,
For my part, now the article meets the standards to be published. However, in lines 111-113 an aspect related to the BMI still appears as a secondary objective.
In the references section, numbers 6, 8 and 36 have the abbreviated name of the journal. And finally, reference 37 has the name "Retos" but in other references (i.e. 33, 43) the journal appears under another name.
And finally, the word "its" is repeated in the title.
Thank you very much for the corrections.
All the best.
Author Response
Dear authors,
For my part, now the article meets the standards to be published. However, in lines 111-113 an aspect related to the BMI still appears as a secondary objective.
In the references section, numbers 6, 8 and 36 have the abbreviated name of the journal. And finally, reference 37 has the name "Retos" but in other references (i.e. 33, 43) the journal appears under another name.
Authors’ response: You have right. Already corrected. Thank you a lot.
And finally, the word "its" is repeated in the title.
Authors’ response: Thank you very much. It was modified.
Thank you very much for the corrections.
All the best.
Authors’ response: Thank you very much for everything.